# Effects of intraocular pressure change on intraocular lens power calculation in primary open-angle glaucoma and ocular hypertension

**Chareenun Chirapapaisan**[☉]**, Akarawit Eiamsamarng**[iD]**\*, Niphon Chirapapaisan**[☉]**, Wara Raksong**[‡]**, Darin Sakiyalak**[‡]**, Sunisa Koodkaew**[‡]**, Audcharawadee Subunnasenee**[‡]

Department of Ophthalmology, Faculty of Medicine Siriraj Hospital, Mahidol University, Bangkok, Thailand

☉ These authors contributed equally to this work.
‡ WR, DS, SK and AS also contributed equally to this work.
\* akarawit.eam@mahidol.ac.th

**Data Availability Statement:** All relevant data are within the manuscript and its Supporting Information files.

## Abstract

This study aimed to assess the effect of intraocular pressure (IOP) changes on biometry and intraocular lens (IOL) power calculation in patients diagnosed with primary open-angle glaucoma (POAG) and ocular hypertension (OHT). This prospective non-randomized cohort study enrolled patients with diagnosed POAG and OHT, presenting with IOP levels exceeding 25 mmHg. Thai Clinical Trials Registry number was TCTR20180912007. Optical biometry, encompassing measurements such as corneal thickness (CCT), keratometry, anterior chamber depth (ACD), and axial length, was conducted before and after IOP reduction. The IOL power was also determined using the SRK/T formula. The main outcomes measured were alterations in biometry and IOL power. Correlations between IOP, biometric parameters, and IOL power were analyzed. In total, 28 eyes were included in the study, with a mean patient age of 65.71±10.2 years. After IOP reduction, all biometric parameters, except CCT and ACD, exhibited a decrease without reaching statistical significance (all p>0.05). Meanwhile, IOL power showed a slight increase of 0.214±0.42 diopters (P = 0.035). The correlation between IOP and biometric parameters was found to be weak. However, there was a moderate correlation between IOP and IOL power ($r^2$ = 0.267). Notably, IOL power tended to increase by more than 0.5 diopters when IOP decreased by more than 10 mmHg (p < 0.001). In conclusion, changes in IOP among patients with POAG and OHT do not significantly impact biometry and IOL power calculations. Nonetheless, it may be prudent to consider a slight adjustment in IOL power when IOP is lowered by more than 10 mmHg.

## Introduction

Biometry plays a crucial role in ensuring the success of cataract surgery. Advanced technology has allowed for accurate measurements, including calculations of intraocular lens (IOL) power. However, there have been reports of unexpected refractive outcomes following cataract

**Funding:** This research was supported by the Research Improvement Fund of the Faculty of Medicine Siriraj Hospital, Mahidol University, Bangkok, Thailand. (R016031039) awarded to CC.

**Competing interests:** The authors have declared that no competing interests exist.

surgery, particularly in cases where biometry measurements were inconsistent or inaccurate, leading to improper IOL power determination [1]. Typically, the newer IOL formulas rely on several biometric parameters: central corneal thickness (CCT), keratometry (K), anterior chamber depth (ACD), lens thickness, and axial length (AL). Certain ocular pathologies can induce changes in ocular dimensions, resulting in deviations in biometric measurements from their normal values. Prior research has shown that elevated intraocular pressure (IOP) in patients with or without glaucoma can alter ocular biometry and potentially affect IOL calculations [2, 3].

The incidence of cataracts and glaucoma, particularly primary open-angle glaucoma (POAG), increases with age. Ocular hypertension (OHT) is considered a major risk factor for the development of POAG [4]. Clinically, many patients present with cataracts and coexisting POAG or OHT. The management of these patients is based on the severity of the diseases. Most ophthalmologists aim to lower IOP to near-normal levels before cataract surgery [5], using either antiglaucoma medications or glaucoma surgery if necessary. In cases where medical treatment fails, surgical drainage of aqueous humor to bypass the trabecular pathway may be considered to achieve sufficient IOP reduction. Previous studies have shown that a significant decrease in IOP following trabeculectomy leads to change in ocular biometry, including ACD and AL, which persist for up to 5 years [6–8]. These findings suggest that substantial changes in IOP can impact biometry. Fortunately, most glaucoma patients respond well to medications [9].

Several publications have suggested that early phacoemulsification is a favorable option for cataract patients with comorbid POAG and OHT. Such a procedure could not only address the cataract issue but also aid in reducing IOP [10, 11]. However, this raises concerns about the impact of IOP changes on cataract surgery outcomes. As mentioned earlier, changes in IOP can affect biometric measurements. The IOL power calculated when the IOP is high before cataract surgery may not be accurate, leading to suboptimal refractive outcomes. Therefore, we conducted this study to assess the influence of IOP on biometric measurements and the IOL power obtained from a standard optical biometer in patients with POAG and OHT. We aimed to compare the findings before and after the reduction in IOP.

## Materials and methods

This study was approved by the Siriraj Institutional Review Board, Faculty of Medicine Siriraj Hospital, Mahidol University, Thailand, and registered with the Thai Clinical Trials Registry (TCTR20180912007). All study procedures adhered to the principles outlined in the Declaration of Helsinki, and written informed consent was obtained from all participants before the study commenced.

This prospective nonrandomized cohort study was conducted at the outpatient clinic of the Department of Ophthalmology, Siriraj Hospital, Mahidol University, Thailand, between April 2017 and March 2018. The inclusion criteria were as follows: (1) patients aged over 20 years diagnosed with POAG or OHT by a glaucoma specialist (D.S.); (2) diagnosis determined based on slit-lamp examination, gonioscopy, optic disc appearance, and visual field test; and (3) IOP levels exceeding 25 mmHg as measured by the standard Goldman applanation technique. Subjects were excluded if they had corneal and/or ocular surface diseases that could affect IOP measurement, K, or refractive status; had a history of ophthalmic surgery; presented with a dense cataract obstructing AL measurement and IOL computation; or were unable to cooperate during the measurements.

All enrolled patients underwent biometry and CCT measurements before undergoing a comprehensive eye examination and initiating antiglaucoma treatment. Biometric

measurements, including K, ACD, and AL, were obtained using a standard time-domain optical coherence interferometry-based optical biometer (IOLMaster 500, Carl Zeiss Meditec AG, Jena, Germany) by a single operator. The IOL power was then calculated using the Sanders–Retzlaff–Kraff/Theoretical (SRK/T) formula for a single-piece foldable IOL (Alcon SA60WF), with an A-constant of 119.0 provided by the IOLMaster. The refractive target was set at emmetropia. Subsequently, CCT was measured using a noncontact air puff tonometer (CT-80 Noncontact Computerized Tonometer, Topcon Corp, Tokyo, Japan).

Following the collection of all parameters, patients underwent a complete ophthalmologic examination. The IOP was confirmed using Goldmann applanation tonometry, consistent with the predefined criteria. Antiglaucoma medications were then prescribed based on the judgment of the glaucoma specialist. Once the IOP was stabilized at or below 20 mmHg, the patients underwent biometry and CCT measurements with the same technician. The interval between the first and second measurements for all subjects was mandated to be within 6 months. Data from longer periods were excluded.

## Statistical analysis

All data were recorded in an Excel 2013 spreadsheet (Microsoft Corporation, Seattle, WA, USA) and analyzed using IBM SPSS Statistics, version 23 (IBM Corp, Armonk, NY, USA). The IOL power, CCT, K, ACD, and AL values are presented as the mean ± standard deviation. Estimation accuracy was evaluated using a 95% confidence interval. Generalized estimating equation (GEE) were performed for all comparisons, with statistical significance defined as a *P* value of less than 0.05. The correlation between the IOP and IOL change was analyzed by Pearson correlation coefficient. A receiver operating characteristic curve (ROC curve) was employed to determine the threshold of IOP change that impacts IOL power.

The reliability of the measurements was assessed using intraclass correlation coefficients. Coefficient values range from 0 to 1, with values above 0.7 signifying acceptable agreement, 0.8 to 0.9 indicating good agreement, and above 0.9 denoting excellent agreement between measurements [12].

## Results

A total of 28 eyes from 18 patients diagnosed with POAG and OHT were included in this study. The mean age of the study population was 65.71±10.2 years, ranging from 50 to 89 years. The demographic characteristics of all patients are detailed in **Table 1**. All patients

**Table 1. Demographic data of patients with POAG and OHT.**

| CHARACTERISTICS | VALUES |
|---|---|
| N | 28 |
| AGE (YEARS) | |
| MIN, MAX | 50, 89 |
| MEAN ± SD | 65.71 ± 10.2 |
| SEX, N (%) | |
| MALE | 19 (67.9) |
| FEMALE | 9 (32.1) |
| DIAGNOSIS | |
| POAG | 12 (42.9) |
| OHT | 16 (57.1) |

Abbreviation: POAG = primary open angle glaucoma; OHT = ocular hypertension; min = minimum; max = maximum

**Table 2. Ocular biometry before and after receiving the anti-glaucoma treatment.**

| Parameters | Mean ± SD | | Mean difference | 95% CI | P-value |
|---|---|---|---|---|---|
| | before receiving anti-glaucoma treatment | after receiving anti-glaucoma treatment | | | |
| CCT (uM) | 549.85±29.08 | 553.53±31.02 | 3.678 | -0.576, 7.933 | 0.090 |
| Flattest K (K1) | 44.45±1.78 | 44.31±1.83 | -0.147 | -0.576, 0.282 | 0.501 |
| Steepest K (K2) | 45.12±1.66 | 45.1±1.71 | -0.020 | -0.234, 0.194 | 0.855 |
| AL (mm) | 23.41±1.04 | 23.38±1.01 | -0.035 | -0.125, 0.055 | 0.446 |
| ACD (mm) | 3.13±0.5 | 3.14±0.55 | 0.011 | -0.092, 0.114 | 0.839 |
| IOL power (D) | 20.37±2.62 | 20.58±2.8 | 0.214 | 0.020, 0.644 | 0.035 |

Abbreviations: CCT central corneal thickness, K keratometry, AL axial length, ACD anterior chamber depth, IOL intraocular lens, uM micrometer, mm milimeter, D dioptor

received treatment with antiglaucoma medications and were scheduled for follow-up visits. The mean IOP before and after treatment with antiglaucoma medications was 28.64±3.31 mmHg and 17.17±2.12 mmHg, respectively. The change in IOP after treatment was statistically significant (P<0.001), with a mean reduction of 11.46 ± 4.12 mmHg. **Table 2** summarizes the IOL power and biometric parameters measured before and after treatment.

The measurement of IOL power showed a statistically significant difference after lowering the IOP to 20 mmHg or below, as obtained by the IOLMaster (IOL power change, 0.214±0.42 D; *P* = 0.035). However, no significant changes were observed in other parameters after lowering the IOP.

Pearson analysis revealed a moderate correlation ($r^2$ = 0.267) between the change in IOL power and the change in IOP (**Fig 1**). An area under the curve of 0.905 (95% CI: 0.79–1.00) indicated a significant cut-off point between a reduction in IOP greater than 10 mmHg and a shift in IOL power of 0.5 diopters. The association of the IOP over 10 and the IOL power change (0.5 diopter) was significant (p < 0.001). The odd ratio was 27.0 (2.5, 291.2).

## Discussion

Accurate preoperative IOL power calculations are crucial for achieving the desired refractive outcomes in cataract surgery. Patients with POAG and OHT often develop cataracts, and high IOP is a modifiable risk factor in these individuals [13]. This study aimed to assess the impact of IOP reduction on IOL power calculations and various biometric parameters, including AL, steepest and flattest K, and ACD. In this study, all subjects had an AL exceeding 22 mm, and the SRK/T formula provided accurate results for this average AL. The SRK/T formula, introduced in 1990 by Retzlaff, Kraff, and Sanders, is considered a third-generation formula developed explicitly for AL values greater than 22 mm. [14]. Moreover, we have used SRK/T formula in our regular practice and already had a good IOL optimization. By opting an IOP threshold of 25 mmHg, the substantial IOP changes are potential to be clinically meaningful.

Optical biometers have gained popularity and are widely used for IOL measurements. Their accuracy is comparable to that of high-precision immersion ultrasound [15]. In our study, we utilized the IOLMaster5 to measure biometric parameters. This approach avoided the globe indentation and deformation that may occur with contact ultrasonic methods, which tend to underestimate AL. Additionally, all biometric measurements were obtained by the same operator (S.K.) to control for technical factors. Our use of a noncontact optical biometer with a reliability check in age-matched patients enhanced the robustness of our study.

We observed a statistically significant increase in SRK/T-calculated emmetropia IOL power after IOP reduction (0.214±0.42 D; P = 0.035). However, there were no significant changes in

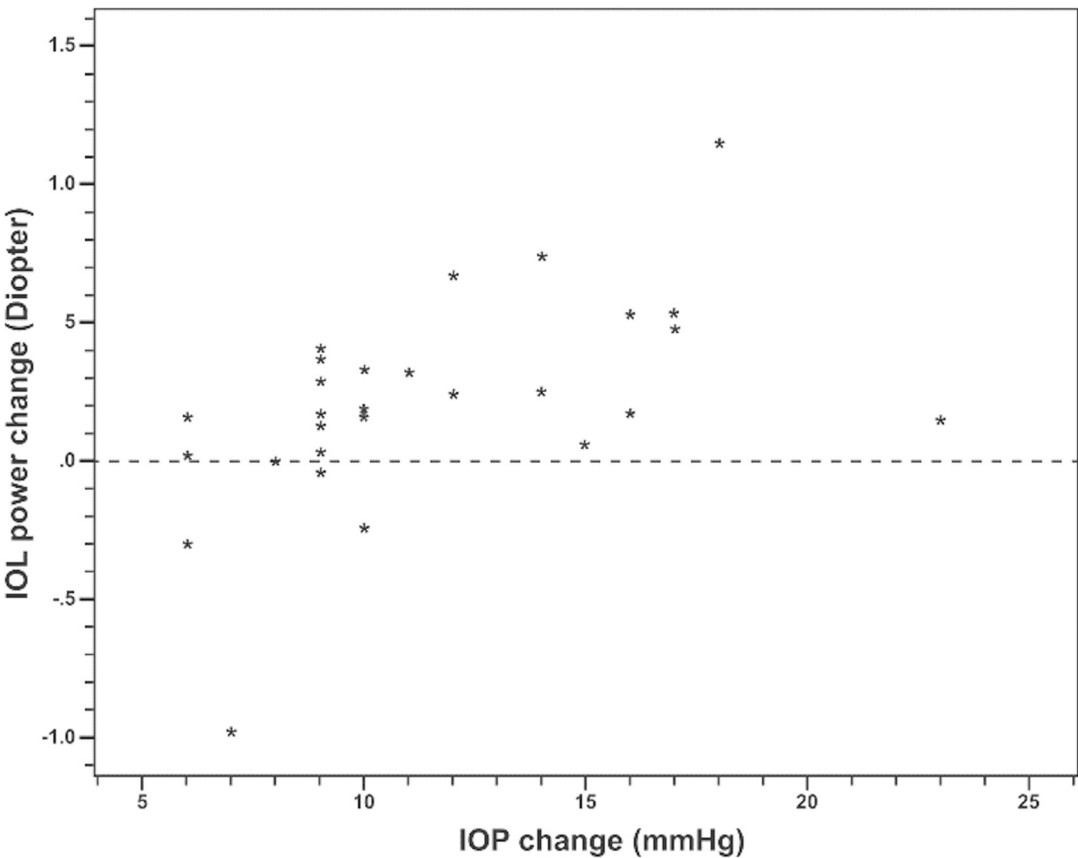

**Fig 1. Scatter plot of correlation between the IOL power change and IOP change.**

AL, steepest and flattest K, or K. The correlation between the change in IOP and the change in IOL power was good inverse ($r^2 = 0.267$). This shift might be attributable to a reduction in AL following the decrease in IOP. Moreover, an IOP drop exceeding 10 mmHg was found to correspond with a 0.5 diopter shift in IOL power.

Husain et al [6]. conducted a study on patients with POAG and PACG who underwent trabeculectomy to lower IOP. They observed a decrease in both ACD and AL following surgery, and these changes persisted for up to 5 years. Similarly, Francis et al [16]. reported a reduction in AL: -0.15 mm at 3 months after trabeculectomy and glaucoma drainage device surgery, as measured by the IOLMaster.

In our study, we also observed a slight reduction in AL after the decrease in IOP, although the difference was not statistically significant (-0.035±0.206 mm; $P = 0.446$). This finding aligns with the proposed mechanism of AL reduction after trabeculectomy, which involves an increase in choroidal and ocular wall thickness associated with the lowering of IOP [17, 18]. Accurate AL measurements are crucial for precise IOL power calculation. Cruysberg et al. [19] reported that AL changes of 0.03 mm and 0.08 mm corresponded to refractive prediction errors of 0.08–0.13 D and 0.20–0.34 D, respectively. As the AL decreases, there is an escalation in refractive prediction error, resulting in a hyperopic shift.

In contrast to AL, changes in ACD are minor and transient, and they do not appear to be influenced by a decrease in IOP. Some studies have observed a temporary shallowing of the ACD following trabeculectomy, but the ACD tends to revert to its presurgical depth within the second week after the operation [20, 21]. Karasheva et al. [22] also demonstrated that a decline

in IOP following filtration surgery does not correlate with a shallowing anterior chamber, as measured by the IOLMaster.

Our study has several limitations that should be acknowledged. First, the small number of study subjects may limit the generalizability of our findings. Additionally, we did not conduct subgroup analysis based on AL, which means that the SRK/T formula may not be suitable for individuals with short AL (<22.00 mm) or extremely long AL. Further research with a larger sample size is needed to validate our results. It is also recommended to utilize newer optical biometers with high repeatability and reliability to investigate the impact of IOP on IOL power using different IOL formulas.

Despite these limitations, the effect size of the differences in parameters was small, which enhances the precision of IOL power calculations. For optimal visual results, it is advisable to measure biometric parameters using noncontact optical biometry for IOL power calculation after stabilizing and effectively controlling IOP.

In conclusion, changes in IOP have a significant impact on IOL power computations. Lowering IOP through antiglaucoma medication treatment leads to a significant increase in emmetropia IOL power. Surgeons should consider the postoperative hyperopic shift of at least 0.5 diopter in patients with an IOP reduction of $\geq 10$ mmHg during cataract surgery.

## Supporting information

**S1 Table. Raw data from all participants.** OHT ocular hypertension, POAG primary open angle glaucoma, IOP Pre intraocular pressure before IOP reduction, IOP Post intraocular pressure after IOP reduction, IOL Power Pre intraocular lens power before IOP reduction, IOL Power Post intraocular lens power after IOP reduction, AL Pre axial length before IOP reduction, AL Post axial length after IOP reduction, ACD Pre anterior chamber depth before IOP reduction, ACD Post anterior chamber depth after IOP reduction, CCT Pre central corneal thickness before IOP reduction, CCT Post central corneal thickness after IOP reduction, K1 Pre flattest keratometry before IOP reduction, K1 Post flattest keratometry after IOP reduction, K2 Pre steepest keratometry before IOP reduction, K2 Post steepest keratometry after IOP reduction, D diopters.
(DOCX)

## Acknowledgments

The authors gratefully acknowledge the editing of this paper by Mr. David Park.

## Author Contributions

**Conceptualization:** Chareenun Chirapapaisan, Akarawit Eiamsamarng, Niphon Chirapapaisan, Wara Raksong, Darin Sakiyalak, Sunisa Koodkaew, Audcharawadee Subunnasenee.

**Data curation:** Sunisa Koodkaew, Audcharawadee Subunnasenee.

**Formal analysis:** Wara Raksong.

**Funding acquisition:** Niphon Chirapapaisan.

**Methodology:** Akarawit Eiamsamarng, Wara Raksong, Darin Sakiyalak.

**Project administration:** Akarawit Eiamsamarng.

**Supervision:** Niphon Chirapapaisan.

**Writing – original draft:** Chareenun Chirapapaisan.

**Writing – review & editing:** Darin Sakiyalak, Sunisa Koodkaew, Audcharawadee Subunnasenee.

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
