## [Decision Letter · Decision Letter 0]

21 Feb 2024

PONE-D-24-02585Effects of intraocular pressure change on intraocular lens power calculation in primary open-angle glaucoma and ocular hypertension.PLOS ONE

Dear Dr. Eiamsamang,

Thank you for submitting your manuscript to PLOS ONE. After careful consideration, we feel that it has merit but does not fully meet PLOS ONE’s publication criteria as it currently stands. Therefore, we invite you to submit a revised version of the manuscript that addresses the points raised during the review process.

**ACADEMIC EDITOR: **Please justify the inclusion of both eyes of some participants rather than randomisation of one eye only.Please justify the use of the cutoff IOP of 25 mmHg (instead of 21 mmHg) for study inclusion.Was there a statistically significant reduction of IOP after medical treatment? Please provide p values.

We look forward to receiving your revised manuscript.

Kind regards,

Nader Hussien Lotfy Bayoumi, M.D., FRCS (Glasgow)

Academic Editor

PLOS ONE

Journal Requirements:

2. In the online submission form you indicate that your data is not available for proprietary reasons and have provided a contact point for accessing this data. Please note that your current contact point is a co-author on this manuscript. According to our Data Policy, the contact point must not be an author on the manuscript and must be an institutional contact, ideally not an individual. Please revise your data statement to a non-author institutional point of contact, such as a data access or ethics committee, and send this to us via return email. Please also include contact information for the third party organization, and please include the full citation of where the data can be found.

3. Please be informed that funding information should not appear in the Acknowledgments section or other areas of your manuscript. We will only publish funding information present in the Funding Statement section of the online submission form. Please remove any funding-related text from the manuscript.

Reviewers' comments:

Reviewer's Responses to Questions

**Comments to the Author**

1. Is the manuscript technically sound, and do the data support the conclusions?

Reviewer #1: No

Reviewer #2: Yes

Reviewer #3: Partly

2. Has the statistical analysis been performed appropriately and rigorously? 

Reviewer #1: No

Reviewer #2: Yes

Reviewer #3: I Don't Know

3. Have the authors made all data underlying the findings in their manuscript fully available?

Reviewer #1: No

Reviewer #2: Yes

Reviewer #3: No

4. Is the manuscript presented in an intelligible fashion and written in standard English?

Reviewer #1: No

Reviewer #2: Yes

Reviewer #3: Yes

5. Review Comments to the Author

Reviewer #1: In general, the process of calculating the power of the intraocular lens and cataract surgery for patients with high eye pressure is not done until the eye pressure is controlled and stabilized for several reasons, some of which are mentioned in this article, so maybe the goal From this study, it contains only a few exceptions and exceptions, and it does not seem practical

As you mentioned, eye biometric changes with eye pressure changes have been mentioned in numerous articles. What is the novelty of the article's message?

Why srkt formula?

While the cct and acd index are not directly used in this formula, and one of your goals is to check the changes of these biometric indices with changes in eye pressure and then calculated changes in iol power.

How is the correlation of two eyes considered?

A change of 0.2 in the mean dif of lens power cannot have an effect on the clinical results

According to the variables that you checked, as well as the reduction in the use of 3rd generation formulas and the widespread use of modern formulas or artificial intelligence, it would be better if you used these formulas, which are better in terms of accuracy in calculating the lens in different AL values.

Many times since 2000 with the advent of optical biometrics, these points have been reviewed and mentioned in various studies.

This amount of error in ref can be different depending on other eye biometric characteristics

Reviewer #2: In this study, the authors evaluated the effects of intraocular pressure change on intraocular lens power calculation in primary open-angle glaucoma and ocular hypertension and observed a nonsignificant reduction in some biometric parameters.

Minor revisions are needed to improve this well-conducted study.

The change in IOL power is different between abstract and text. “IOL power showed a slight increase of 0.213±0.54 diopters (P=0.076) vs (IOL power change, 0.213±0.377 D; P=0.006)”

Figure Legends for Fig 1 and Fig 2 seem to be interchanged.

Reviewer #3: • Some statement in the “Introduction” need to be supported by references: “Most ophthalmologists aim to lower IOP to near-normal levels before cataract surgery, using either antiglaucoma medications or glaucoma surgery if necessary”, “Fortunately, most glaucoma patients respond well to medications”.

• The number of eyes involved in the study needed a statistically justified sample size calculation. The inclusion of both eyes of the same patient in the study also needs statistical justification.

• In the abstract, the statement “IOL power showed a slight increase of 0.213±0.54 diopters (P=0.076)” is not accurate as p=0.006 in table 2.

• Figure 2 and its legend are not clear.

• The control group mentioned at the end of the results has no place. If it is a case-control study, this should be considered in the methodology and the data of the control group should be presented in tables in comparison with the study group.

• These statements in the discussion are not supported in the results: “more hyperopic shift in patients whose IOP was reduced by ≥8 mmHg” and “IOP drop exceeding 10 mmHg was found to correspond with a 0.5 diopter shift in IOL power”.

• The second part of the conclusion is not supported by the results.

6. PLOS authors have the option to publish the peer review history of their article (what does this mean?). If published, this will include your full peer review and any attached files.

Reviewer #1: No

Reviewer #2: No

Reviewer #3: No

---

## [Author Response · Author response to Decision Letter 0]

4 May 2024

Dear editor and reviewers,

We appreciate you taking the time to review this article. Thank you very much indeed for your thoughtful comments. 

We has corrected Data availability statement in a system to The datasets generated and analyzed during the study are available from the corresponding author upon reasonable request that related with in manuscript. 

Response to editor.

1. Please justify the inclusion of both eyes of some participants rather than randomisation of one eye only.

Response:

According to the uneven incident of OHT and POAG with IOP higher than 25 mmHg particitated in the current study, eyes which met the inclusion criteria were recruited. However, we have used Generalized Estimating Equation (GEE) for the statistical analysis to compensate the incorporation of data derived from bilateral eyes and unilateral eye participants. 

We changed a sentence in statistical analysis (page 8, line135-139). “Generalized estimating equation (GEE) were performed for all comparisons, with statistical significance defined as a P value of less than 0.05.”

The results were the same after changing the method of statistical analysis. We have already revised the results in table 2 as below. 

Table 2 Ocular biometry before and after receiving the anti-glaucoma treatment

Parameters Mean ± SD

 Mean difference 95% CI P-value

 before receiving anti-glaucoma treatment after receiving anti-glaucoma treatment 

CCT (uM) 549.85±29.08 553.53±31.02 3.678 -0.576, 7.933 0.090

Flattest K (K1) 44.45±1.78 44.31±1.83 -0.147 -0.576, 0.282 0.501

Steepest K (K2) 45.12±1.66 45.1±1.71 -0.020 -0.234, 0.194 0.855

AL (mm) 23.41±1.04 23.38±1.01 -0.035 -0.125, 0.055 0.446

ACD (mm) 3.13±0.5 3.14±0.55 0.011 -0.092, 0.114 0.839

IOL power (D) 20.37±2.62 20.58±2.8 0.214 0.020, 0.644 0.035

Abbreviations: CCT central corneal thickness, K keratometry, AL axial length, ACD anterior chamber depth, IOL intraocular lens, uM micrometer, mm milimeter, D dioptor

……………………………………………………………………………………………………

2. Please justify the use of the cutoff IOP of 25 mmHg (instead of 21 mmHg) for study inclusion.

Response:

In clinical practice, patients with ocular hypertension (OHT) are often untreated if their intraocular pressure (IOP) are below 25 mmHg. Mild reduction of IOP may not significantly affect the ocular biometry. By opting an IOP threshold of 25 mmHg, the substantial IOP changes are potential to be clinically meaningful. This approach reduces data noise and enhances the statistical power of the analysis.

We added a sentence in discussion.

“By opting an IOP threshold of 25 mmHg, the substantial IOP changes are potential to be clinically meaningful.” (page 10, line176-177)

………………………………………………………………………………………………………

3. Was there a statistically significant reduction of IOP after medical treatment? Please provide p values.

Response:

Yes, there was. The change in IOP after treatment was statistically significant (P<0.001), with a mean reduction of 11.46 ± 4.12 mmHg. We added a p-value in the sentence. Page 8 line 150.

………………………………………………………………………………………………………

Response to reviewer #1 .

1. In general, the process of calculating the power of the intraocular lens and cataract surgery for patients with high eye pressure is not done until the eye pressure is controlled and stabilized for several reasons, some of which are mentioned in this article, so maybe the goal From this study, it contains only a few exceptions and exceptions, and it does not seem practical

As you mentioned, eye biometric changes with eye pressure changes have been mentioned in numerous articles. What is the novelty of the article's message?

Response:

Up to present, there has been no study analyzed the effect of IOP changes on intraocular lens (IOL) power. In the current study, we emphasize this effect and point out the magnitude of IOL power changes after significant IOP reduction. This provides accurate lens values, resulting in good vision after cataract surgery.

………………………………………………………………………………………………………

2. Why srkt formula?

While the cct and acd index are not directly used in this formula, and one of your goals is to check the changes of these biometric indices with changes in eye pressure and then calculated changes in iol power.

Response:

The axial lengths of our patients were in average range. Any IOL formulas should yield accurate results. Moreover, we have used SRK/T formula in our practice and already had a good IOL optimization.

We added a sentence in discussion. 

“Moreover, we have used SRK/T formula in our regular practice and already had a good IOL optimization.” (page 10, line 175)

………………………………………………………………………………………………………

3. How is the correlation of two eyes considered?

Response:

We use generalized Estimating Equation (GEE) for the incorporation of the correlation between the two eyes of the same participant. 

We changed a sentence in statistical analysis (page 8, line135-139). “Generalized estimating equation (GEE) were performed for all comparisons, with statistical significance defined as a P value of less than 0.05.”

The results were the same after changing the method of statistical analysis. We have already revised the results in table 2.

………………………………………………………………………………………………………

4. A change of 0.2 in the mean dif of lens power cannot have an effect on the clinical results

According to the variables that you checked, as well as the reduction in the use of 3rd generation formulas and the widespread use of modern formulas or artificial intelligence, it would be better if you used these formulas, which are better in terms of accuracy in calculating the lens in different AL values.

Many times since 2000 with the advent of optical biometrics, these points have been reviewed and mentioned in various studies.

This amount of error in ref can be different depending on other eye biometric characteristics

Response:

Thank you for your point of view. It is true that the modern IOL formulas may yield more accurate results. As we mentioned earlier, the axial lengths of our patients were in average range. Any IOL formulas should yield similar results. Moreover, we have used SRK/T formula in our practice and already had a good IOL optimization.

We added a sentence in discussion. 

“Moreover, we have used SRK/T formula in our regular practice and already had a good IOL optimization.” (page 10, line 175)

………………………………………………………………………………………………………

Response to reviewer # 2.

Reviewer #2: In this study, the authors evaluated the effects of intraocular pressure change on intraocular lens power calculation in primary open-angle glaucoma and ocular hypertension and observed a nonsignificant reduction in some biometric parameters.

Minor revisions are needed to improve this well-conducted study.

1. The change in IOL power is different between abstract and text. “IOL power showed a slight increase of 0.213±0.54 diopters (P=0.076) vs (IOL power change, 0.213±0.377 D; P=0.006)”

Response:

Thank you for your kindness and sorry for the confusion. 

We use generalized Estimating Equation (GEE) for the incorporation of the correlation between the two eyes of the same participant. So IOL power showed a slight increase of0.214±0.42 diopters (P=0.035). 

We corrected them in abstract, text and table 2.

Page 3 line 57. Meanwhile, IOL power showed a slight increase of 0.214±0.42 diopters (P=0.035). 

Page 10 line 158. (IOL power change, 0.214±0.377 D; P=0.035).

………………………………………………………………………………………………………

2. Figure Legends for Fig 1 and Fig 2 seem to be interchanged.

Response:

The legend of Figure 1 is “Fig 1. Scatter plot of correlation of IOL power change and IOP change.” Page 10, line 165.

We decided to delete Figure 2. 

………………………………………………………………………………………………………

Response to reviewer #3.

Reviewer #3: 

1• Some statement in the “Introduction” need to be supported by references: “Most ophthalmologists aim to lower IOP to near-normal levels before cataract surgery, using either antiglaucoma medications or glaucoma surgery if necessary”, “Fortunately, most glaucoma patients respond well to medications”.

Response:

Thank you very much indeed for your comments.

We have added references to these sentences.

“Most ophthalmologists aim to lower IOP to near-normal levels before cataract surgery[5], using either antiglaucoma medications or glaucoma surgery if necessary” (Page 5, line 82)

5. Heltzer JM. Is cataract surgery recommended on a patient with high IOP?. 2019 [cited 20 Apr 2024]. Available from: https://www.aao.org/eye-health/ask-ophthalmologist-q/is-cataract-surgery-recommended-on-patient-with-hi.

“Fortunately, most glaucoma patients respond well to medications”. [9] (Page 6, line89)

9. Expert Interview - Glaucoma Risk Factors With Andrew G. Iwach, MD. 2012 [20 Apr 2024]. Available from: https://www.healio.com/news/ophthalmology/20120331/expert-interview-glaucoma-risk-factors-with-andrew-g-iwach-md.

………………………………………………………………………………………………………

2• The number of eyes involved in the study needed a statistically justified sample size calculation. 

Response:

This was a pilot study. We did not calculate the sample size.

………………………………………………………………………………………………………

3. The inclusion of both eyes of the same patient in the study also needs statistical justification.

Response:

We use generalized Estimating Equation (GEE) for the incorporation of the correlation between the two eyes of the same participant.

We changed a sentence in statistical analysis (page 8, line135-139). “Generalized estimating equation (GEE) were performed for all comparisons, with statistical significance defined as a P value of less than 0.05.”

The results were the same after changing the method of statistical analysis. We have already revised the results in table 2.

………………………………………………………………………………………………………

4• In the abstract, the statement “IOL power showed a slight increase of 0.213±0.54 diopters (P=0.076)” is not accurate as p=0.006 in table 2.

Response:

We are sorry for the confusion. 

We use generalized Estimating Equation (GEE) for the incorporation of the correlation between the two eyes of the same participant. So IOL power showed a slight increase of0.214±0.42 diopters (P=0.035). 

We corrected them in abstract, text and table 2. 

Page 3 line 57. Meanwhile, IOL power showed a slight increase of 0.214±0.42 diopters (P=0.035). 

Page 10 line 158 (IOL power change, 0.214±0.377 D; P=0.035).

………………………………………………………………………………………………………

5• Figure 2 and its legend are not clear.

Response:

We decided to delete Figure2.

………………………………………………………………………………………………………

6• The control group mentioned at the end of the results has no place. If it is a case-control study, this should be considered in the methodology and the data of the control group should be presented in tables in comparison with the study group.

Response:

We deleted this paragraph as you recommend.

………………………………………………………………………………………………………

7• These statements in the discussion are not supported in the results: “more hyperopic shift in patients whose IOP was reduced by ≥8 mmHg” and “IOP drop exceeding 10 mmHg was found to correspond with a 0.5 diopter shift in IOL power”.

• The second part of the conclusion is not supported by the results.

Response:

We deleted “We noticed a trend toward a more hyperopic shift in patients whose IOP was reduced by ≥8 mmHg”

“IOP drop exceeding 10 mmHg was found to correspond with a 0.5 diopter shift in IOL power”.

• The second part of the conclusion is not supported by the results.

We explained more details in statistic analysis (page 8, line137-139) and results (page 10 , line 160-164)

Page 8, line137-139 

“The correlation between the IOP and IOL change was analyzed by Pearson correlation coefficient. A receiver operating characteristic curve (ROC curve) was employed to determine the threshold of IOP change that impacts IOL power.”

Page 10, line 160-163

“Pearson analysis revealed a moderate correlation (r2=0.267) between the change in IOL power and the change in IOP (Fig 1). An area under the curve of 0.905 (95% CI: 0.79-1.00) indicated a significant cut-off point between a reduction in IOP greater than 10 mmHg and a shift in IOL power of 0.5 diopters. The association of the IOP over 10 and the IOL power change (0.5 diopter) was significant. (p < 0.001) The odd ratio was 27.0 (2.5, 291.2).”

---

## [Editor Report · Decision Letter 1]

8 May 2024

Effects of intraocular pressure change on intraocular lens power calculation in primary open-angle glaucoma and ocular hypertension.

PONE-D-24-02585R1

Dear Dr. Eiamsamarng,

We’re pleased to inform you that your manuscript has been judged scientifically suitable for publication and will be formally accepted for publication once it meets all outstanding technical requirements.

Kind regards,

Nader Hussien Lotfy Bayoumi, M.D., FRCS (Glasgow)

Academic Editor

PLOS ONE

Additional Editor Comments (optional):

Thank you for responding well to all review comments.
---

## [Editor Report · Acceptance letter]

30 May 2024

PONE-D-24-02585R1 

PLOS ONE

Dear Dr. Eiamsamarng, 

I'm pleased to inform you that your manuscript has been deemed suitable for publication in PLOS ONE. Congratulations! Your manuscript is now being handed over to our production team.

Kind regards, 

on behalf of

Professor Nader Hussien Lotfy Bayoumi 

Academic Editor

PLOS ONE